# Elucidating trends and underlying drivers of neonatal mortality stagnation in Nepal: An analytical perspective on the 2016 and 2022 Demographic and Health Surveys

Khem Narayan Pokhrel[1]*, Resham Khatri[2], Suman Sapkota[3], Kalpana Gaulee Pokhrel[1], Gauri Pradhan[4], Tulsi Ram Thapa[4], Kamal Prasad Chapain[1], Thomas Pullum[5], Fern Greenwell[5]

1 Development and Research Service International Nepal, Kathmandu, Nepal, 2 School of Public Health, University of Queensland, Brisbane, Queensland, Australia, 3 Nepal Health Economics Association, Kathmandu, Nepal, 4 Government of Nepal, Ministry of Health and Population, Kathmandu, Nepal, 5 ICF International Inc, Calverton, Maryland, United States of America

* pokkhrelkhemn@gmail.com

## Abstract

### Background

Nepal has made significant progress in reducing the Neonatal Mortality Rate (NMR) over the past two decades. However, since 2016, NMR has stagnated at 21 deaths per 1,000 live births, indicating stalled improvements in neonatal survival. Past studies highlighted the disparities, with socioeconomically disadvantaged groups experiencing a higher rate of neonatal deaths. However, limited evidence exists on NMR trends and determinants in Nepal that examined the factors with the trend. This study analyzed NMR trends and key determinants using data from the two most recent Nepal Demographic and Health Surveys (NDHS).

### Methods

NDHS 2016 and NDHS 2022 data were used to calculate NMR. Both surveys received ethical approval from the Nepal Health Research Council. The study included 106 neonatal deaths out of 5,087 live births in 2016 and 105 out of 5,192 in 2022. Independent variables included household characteristics, parental factors, pregnancy-related factors, maternal and newborn care, women's empowerment, and health system factors. NMRs were constructed using births within completed months from 1 to 61. A general linear model assessed NMR trends, while logit regression identified key determinants.

**Data availability statement:** The data used for this study are available in tables and supporting files here.

**Funding:** The author(s) received no specific funding for this work.

**Competing interests:** The authors have declared that no competing interests exist.

## Results

While national NMR remained unchanged since 2016, an increasing trend was observed among disadvantaged groups and mothers who did not utilize maternal/newborn health services. NMR rose from 27.3 to 27.8 per 1,000 live births (p = 0.001) among poor and poorest households. Similarly, women with no education experienced higher NMR at 29.3% in 2022 compared to 25.7% in 2016 (p = 0.002). Maithili-speaking mothers had higher NMR (27.4 in 2022 vs. 23.4 in 2016, p < 0.001). Women lacking decision-making power in healthcare had higher NMRs of 25.9 in 2022 vs. 23.4 in 2016 (p = 0.021). Women who were not assisted by skilled birth attendants (SBA). had significantly higher NMR compared to those, who were assisted by SBA (p = 0.010).

## Conclusions

Targeted health system interventions are needed for disadvantaged groups covering those who had low education, from poor households, low health care decision making and lack access to SBA assisted delivery. While determinants have been explored, further targeted studies are warranted to uncover the causes of neonatal deaths in Nepal.

## Introduction

The world has achieved a significant reduction in neonatal mortality in recent years. Neonatal deaths have fallen by more than half from 5 million in 1990 to 2.4 million in 2020 [1]. Nonetheless, mortality in the first month has declined more slowly than mortality under age five, simply because the risk of death is most acute during delivery and in the vulnerable hours and days after birth. Nearly half of under-five deaths (47 percent) occur during this precarious period when the newborn transitions to risky conditions in a new environment, precisely when the infant and mother are nearly exhausted from labor and delivery [1]. Even when the numbers of neonatal deaths decline, the share of the deaths among under-five have tended to increase over time, most notably in Low- and middle-income countries (LMICs) where relatively more early childhood deaths are prevented than newborn deaths. Globally, neonatal disorders have ranked sixth as leading cause of all deaths since 2013 [2]. The burden of global newborn deaths is mostly shared between sub-Saharan Africa (43%) and southern Asia (36%) [1].

Recognizing that targeted investments are required to save newborn lives, the Sustainable Development Goals (SDGs) explicitly include an indicator for neonatal mortality: SDG target 3.2 calls for all countries to end preventable deaths of newborns and children aged under five years, aiming to reduce the Neonatal Mortality Rate (NMR) to 12 or fewer deaths per 1,000 live births and the under-five mortality rate (U5MR) to 25 or fewer per 1,000 live births by 2030 [3].

In Nepal, neonatal deaths accounted for 64% percent of under-five mortality in 2020, up from 43% in 2000 [4,5]. Further, maternal and neonatal disorders were sixth

leading cause of deaths in 2019, but progress is being made as they ranked as second leading cause of deaths in 1990 [2]. Verbal autopsy results of the NDHS 2016 have shown that the most common underlying causes of neonatal deaths are respiratory and cardiovascular disorders of the perinatal period (31%) and complications of pregnancy, labor, and delivery (31%), with more than half (56%) of deaths occurring at home [6]. The country needs to persevere in its implementation and expansion of neonatal health interventions to overcome the flattening of the downward trend, which is a key concern and is a major motivation for this study [1]. Despite the increase in coverage of antenatal care (ANC) check-ups—which led to institutional delivery—women from remote areas, with higher birth order, and those of Maithili speaking population were less likely to use delivery and postnatal care (PNC) services and perceived problems when not having access to female providers [7]. As a case in point, Nepal has not observed continuous declines despite the improved utilization of MNH interventions [8].

Despite the Nepal's effort to reduce NMRs such as development of policy and policies and programs with an aim to provide life-saving interventions to infants in their first days and months of life. Nepal has prioritized Maternal and Newborn Health (MNH) care, uncovering various policies and programs to lower excessive maternal and newborn deaths. Fig 1 shows the Government of Nepal (GoN) stepping up to devise policies and programs for newborn care.

Over the last 30 years, Ministry of Health and Population (MoHP) has formulated and implemented several MNH-related initiatives, including integrating family planning and community health [9], developing the National Neonatal Health Strategy (2004) [10] and providing quality promotive, preventive, and curative neonatal health services. The National Safe Motherhood and Newborn Health Long-term Plan 2006–2017 supported the expansion of Comprehensive Emergency Obstetric and Newborn Care (CEmONC) facilities, Basic Emergency Obstetric and Newborn Care (BEmONC) facilities, and health posts with skilled birth attendants (SBAs) [11]. The implementation of the Safe Delivery Incentive Programme (SDIP) in 2005 ensured that delivery attended by SBAs in health institutions with the provision of cash incentives

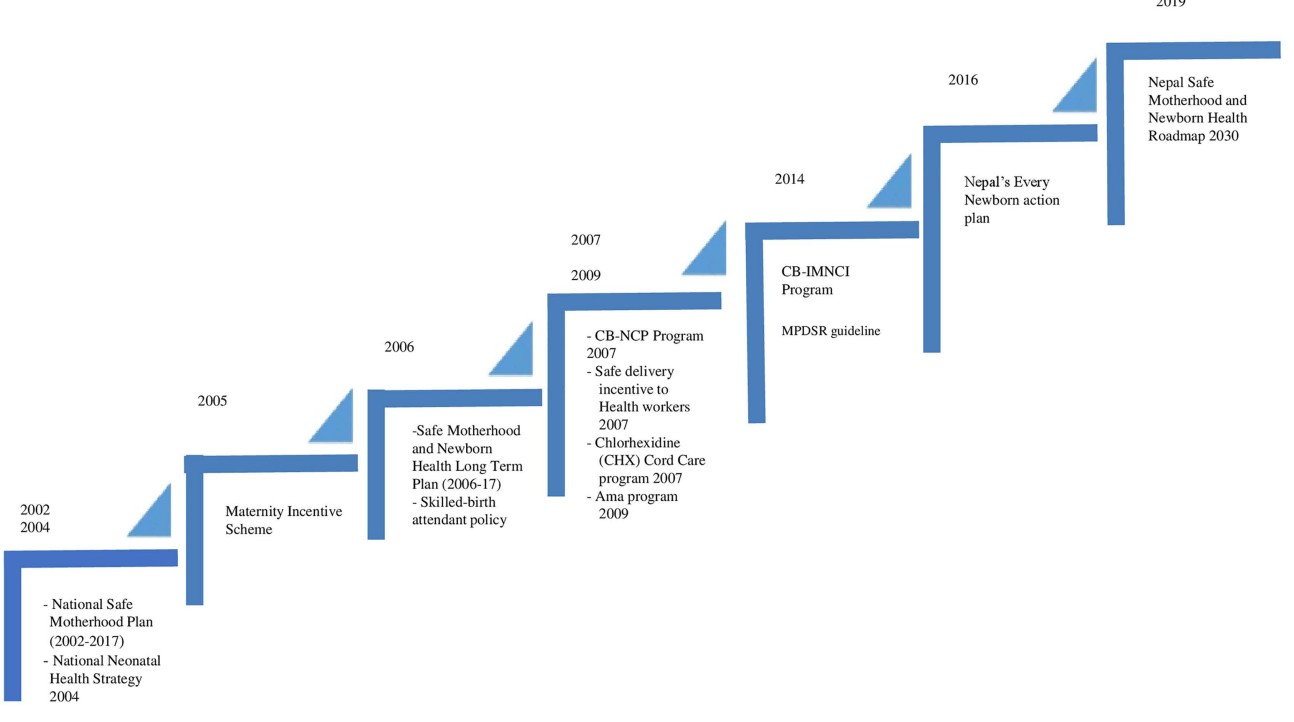

**Fig 1. Policies and programs.**

as transport costs to women who deliver in health institutions, and free delivery services in health facilities (expanded in 2009) [12]. Similarly, Nepal piloted a Community-based Newborn Care Program (CB-NCP) in 2007 and later scaled it up nationwide. The program had four objectives focusing on preventing and managing newborn sepsis, babies with hypothermia, Low Birth Weight (LBW), birth asphyxia, referral of sick newborns; and providing seven essential MNH interventions: Behavior change communication, promoting institutional delivery and clean delivery practices in case of home deliveries, PNC, community case management of possible severe bacterial infection, care of LBW newborns, preventing and managing hypothermia, and managing birth asphyxia [13].

The interventions of CB-NCP were merged into the Integrated Management of Neonatal and Childhood Illness (IMNCI), referring to the new Community-based Integrated Management of Neonatal and Childhood Illness (CB-IMNCI) program [14,15]. Currently, Nepal's newborn care interventions are guided by the Nepal's Every Newborn Action Plan (NENAP) 2016 [16] and the Nepal Safe Motherhood and Newborn Health Roadmap 2030 [17] with the vision of "no preventable deaths of newborns or stillbirths" and targets to reduce NMR to 11 per live births and 13 stillbirths (per 1,000 total births). By 2035, NENAP will emphasise improving equity, delivery, and quality of health care around birth and services and focus on care for small and sick newborns [16]. The Safe Motherhood and Neonatal Health Roadmap 2030 calls for improved governance for MNH services, increased availability of equitable, high-quality MNH services, strengthened emergency preparedness and response, and increased maternal and neonatal care demand.

Despite the myriads of policies and MNH interventions, Nepal has been experiencing challenges in providing quality essential MNH care services. Evidence needs to updated and trends of the neonatal mortality with protective and risk factors over the recent years will provide the timely guidance to the health system of Nepal to accelerate efforts on reduction of neonatal mortality. Therefore, this study analysed the trends and examined the key determinants using data from the two most recent Nepal Demographic and Health Surveys (NDHS).

### Conceptual framework of the study

The conceptual framework is developed using previous studies about the determinants of neonatal mortality, those measured in DHS surveys [18–21]. The determinants are organized into blocks of indicators that are of programmatic importance and also serve to facilitate the analysis objectives. Fig 2 shows that neonatal mortality can be influenced by the variables included in the following blocks: household characteristics, maternal characteristics, women empowerment, mother's participation in health-related programs, Parental characteristics, newborn characterstics, and pregnancy-related characteristics. The variables and their measures are given in S7 Table.

### Methods

This study used data from the 2016 and 2022 NDHS [5,6] to examine the trends and determinants of neonatal mortality in Nepal. The NDHS is a nationally representative household survey. Detailed information for the household survey covering sample designs, sample selection, and sample weighting are available in the final reports of NDHSs [5,6]. Table 1 provides information on the units of analysis from the two household surveys used for this analysis, including births and neonatal deaths in the five years preceding the survey. In 2016, 12,862 women aged 15–49 years were interviewed. Out of the total live births (5,087), 106 neonatal deaths were recorded. In 2022, 14,845 women aged 15–49 years were interviewed. Of the total 5,192 live births, 105 neonatal deaths were recorded.

### Study variables

The outcome variable is NMR, which is a binary variable indicating the death or survival of a newborn during the first month of life. NMR is a key indicator published in the final report of the NDHS and made available through the STATcompiler platform [22]. NDHS calculation for NMR was based on neonatal deaths in the first month of life, or 30 days, which was shown in NDHS final report. However, this study team used the cohort approach and applied the WHO's definition of

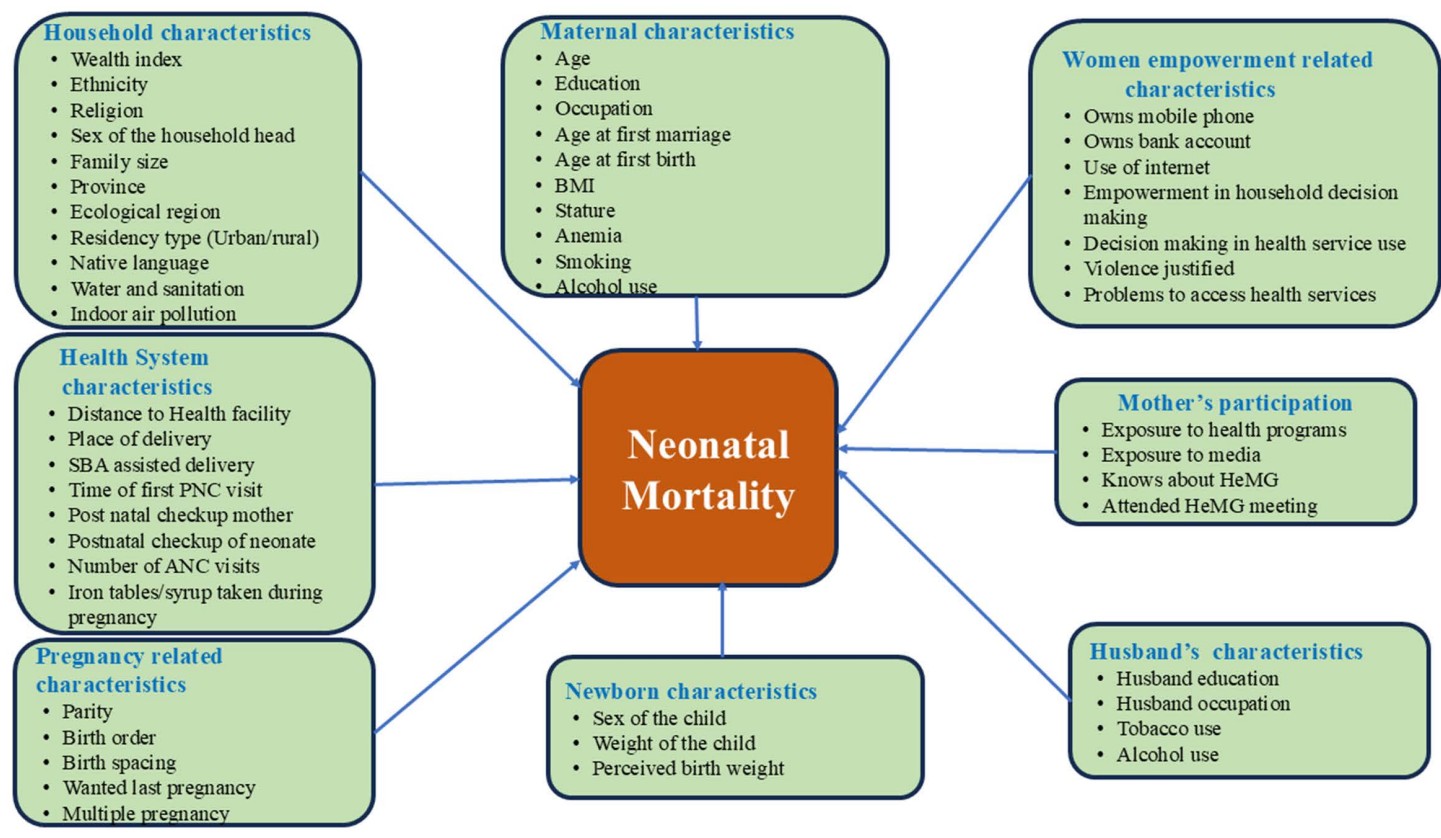

**Fig 2. Conceptual Framework.**

**Table 1. Sampling description of 2016 and 2022 NHDSs.**

| Year | Date of field-work | Reference period for 5-year neona-tal mortality estimates | Refer-ence date | Number of house-holds inter-viewed | House-hold response rate | Number of women age 15–49 inter-viewed | Eligible women response rates | Number of live births, 5 years preceding interview | Number of early neona-tal deaths, 5 years preced-ing interview | Number of late neonatal deaths, 5 years preced-ing interview | Number of neonatal deaths, 5 years preced-ing interview |
|---|---|---|---|---|---|---|---|---|---|---|---|
| 2016 | June 6, 2016 – January 31, 2017 | 2001-2016 | 2013.5 | 11,040 | 98.5 | 12,862 | 98.3 | 5087 | 84 | 22 | 106 |
| 2022 | January 5, 2022 – June 22, 2022 | 2016-2021 | 2019.8 | 13,786 | 99.7 | 14,845 | 97.4 | 5192 | 85 | 20 | 105 |

neonatal death, which is the death of a live birth in the first 28 days (completed days 0–27) [23]. In the report, the rate had been remarkably flat and at the same level in the 5-year period prior to the surveys, at 20.8 deaths per 1,000 live births in both the 2016 and 2022 surveys, interrupting an otherwise smooth decline over previous decades. The calculated NMRs according to WHO NMR definition for Nepal are very close to the NDHS NMR estimates: 20.9 and 20.3 per 1,000 live births for the 2016 and 2022 surveys, respectively.

The study team analyzed 43 independent variables from the 2016 and 2022 NDHSs to estimate their direct or indirect effect on neonatal death. The set of variables are organized into eight groups, as they are presented in the conceptual model (see Fig 2): Household characteristics, maternal characteristics, women empowerment characteristics, mother's participation, husband characteristics, newborn characteristics, pregnancy-related characteristics, and health system characteristics.

### Data analyses

To examine the significant changes in neonatal mortality, the study team calculated the early neonatal mortality rate (ENMR), late neonatal mortality rate (LNMR), and NMR per 1,000 live births for the past five years using the birth histories in the 2016 and 2022 surveys. The ENMR refers to deaths in the first seven days, the NMR to deaths in the first 28 days, and LNMR = NMR − ENMR. The ENMR, LNMR, and NMR are calculated within categories of covariates, which have been grouped as shown in the conceptual framework (see Fig 2). All frequencies and rates are weighted to correct for the complex sample design. The study team used the birth cohort approach. The numbers of births and deaths for the past five years—the numerators and denominators of the rates—and the resulting rates per 1,000 live births for each category for each of the independent variables, are presented for the 2016 survey (S1 Table) and the 2022 survey (S2 Table). The team used a logit model to obtain 95 percent confidence intervals for the estimates. After, the team calculated the difference as the value in the 2022 survey minus the value in the 2016 survey, whereas a positive difference suggests an increase, and a negative difference implies a decline. In addition to 95 percent confidence intervals for the difference, the team calculated p-values to assess the significance of the change for each category of each covariate between 2016 and 2022. S3–S5 Tables present these results for ENMR, LNMR, and NMR, respectively.

S6 Table summarizes S3–S5 Tables of it, showing the significance level, if any, of change in rates. For each p-value, symbols are assigned to facilitate interpretation. The symbols show the direction and strength of change: + for an increase with $p < 0.05$, ++ for an increase with $p < 0.01$, and +++ for an increase with $p < 0.001$. The -, --, and --- symbols represent significant declines in the respective rates. According to the conceptual model, the results are organized into blocks of variables to facilitate interpretation.

The study team also analysed the 2022 NDHS data to identify the subpopulations with the highest and lowest levels of neonatal mortality. We developed a logit regression model for each covariate and obtained the resulting p-values and pseudo-$R^2$ values for the NMR. McKelvey and Zavoina's pseudo-$R^2$ gives the proportion of residual variance that is explained by the covariate. From this analysis, the team identified the categories of the significant ($p < 0.05$) covariates for which the NMR was estimated to be 25 or more, and the categories of the significant ($p < 0.05$) covariates for which the NMR was estimated to be 15 or less. The team assessed the difference in the NMR levels between the 2016 and 2022 NDHSs for this subset of significant covariates. Again, the results were organized into blocks of variables per the conceptual model.

### Ethical considerations

NDHSs obtained ethical approvals from the Nepal Health Research Council and a separate approval was not sought for this study as it is based on secondary analysis of the NDHSs. The study followed the procedures to obtain written consent from the participants and consent from parents or guardians from those who were below 18 years of age. Further, the study team obtained the authorization letter was obtained from ICF, The Demographic and Health Survey to use the data.

### Results

### Trend of NMR

**NMR by day of death from the birth in 2016 and 2022 NDHSs.** The highest number of neonatal deaths occurred on the first day of birth. In 2016, NMR was 11.6 per 1,000 live births on the day of birth, whereas in 2022, the rate was lower

than this at 7.2. ENMR (from birthday to sixth day) accounted for 16.3 per 1,000 live births in 2016 and 16.6 per 1,000 live births in 2022 (Fig 3).

**NMR by provinces within ten years preceding the 2016 and 2022 NDHSs.** NMR varies by province in Nepal. In 2016, Sudurpaschim Province had the highest NMR at 41.4 per 1,000 live births. Lumbini, Karnali, and Madhesh provinces had similar NMR at around 30. Koshi Province had an NMR slightly above the national level (21.6), while Bagmati (16.6) and Gandaki provinces (14.8) had mortality below the national level in 2016. The provincial disparity in terms of NMR was markedly reduced in 2022. Except for Bagmati Province, the rates declined in all provinces. Lumbini, Karnali, Madesh, and Sudurpaschim provinces had similar NMRs, ranging from 24–27. Gandaki province had the lowest mortality at 8.5 per 1,000 live births, whereas Bagmati (18.2) and Koshi provinces (19.9) had similar NMRs (Fig 4).

**NMR according to Residence and Wealth Index, 0–5 years preceding the 2016 and 2022 NDHSs.** The equity gap in neonatal mortality is estimated using the DHS wealth index. The disparity between poor/poorest and richer/richest increased from 2016 to 2022. In 2016, poorer and poorest had an NMR of 27.3, compared to 16.3 for richer and richest. In 2022, the poorest and poorer had an NMR of 27.7, triple the level of 8.7 among richer and richest (Fig 5).

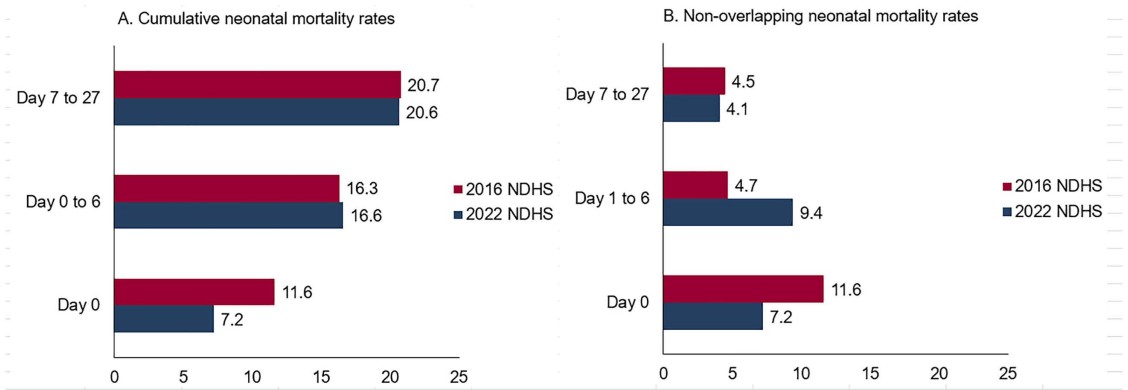

**Fig 3. NMR by time of death from birth in the NDHS 2016 and 2022 surveys.**

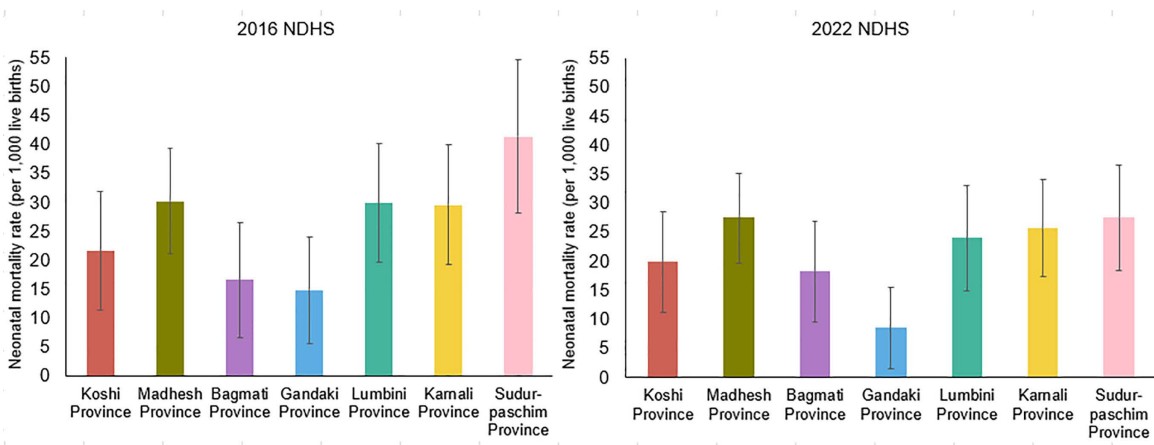

**Fig 4. NMR differentials across provinces, 0–10 years preceding the 2016 and 2022 NDHSs.**

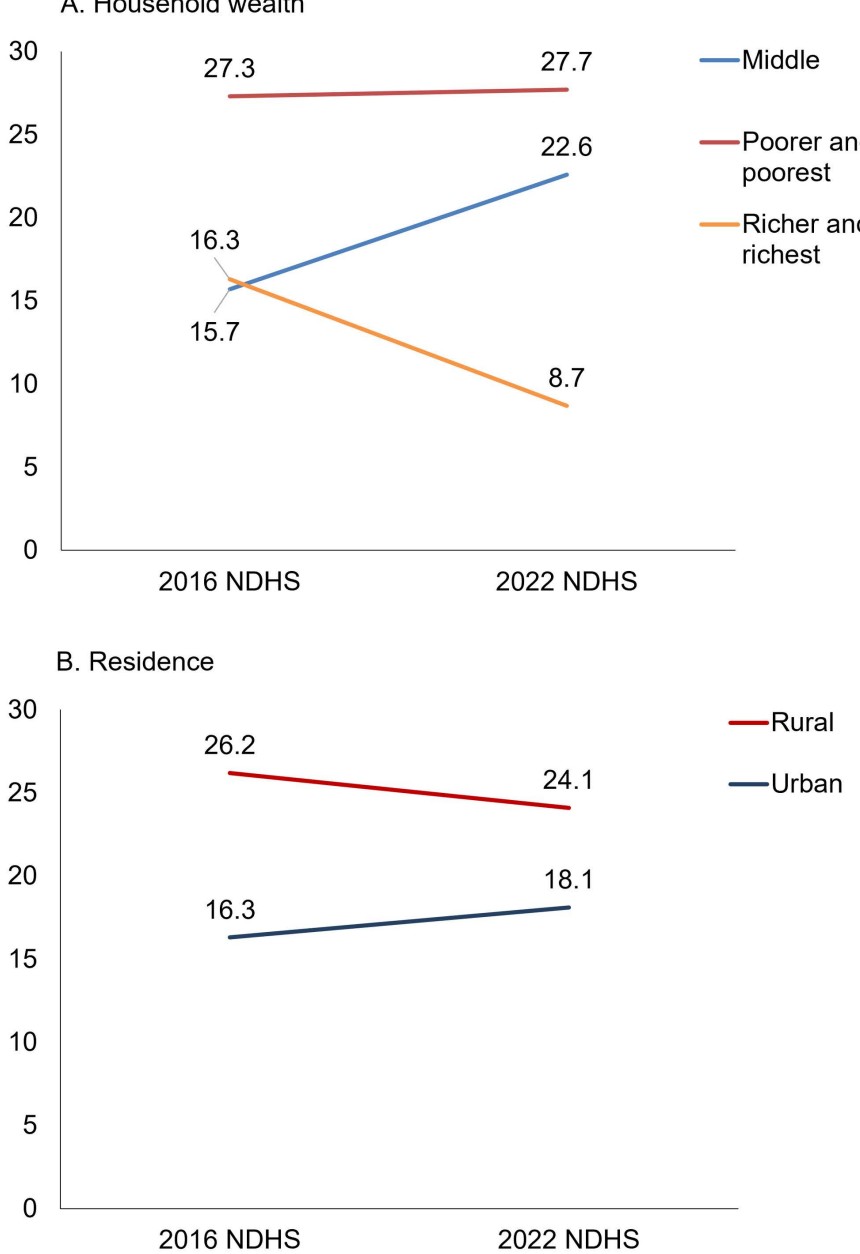

Fig 5. **NMR differentials across residence and wealth index, 0–5 years preceding the 2016 and 2022 NDHSs.**

The disparity in NMR was higher between rural and urban types of residence in 2016 compared to 2022. In 2016, NMR in rural households was 26, compared to NMR at 16 in urban households. In 2022, NMR was slightly higher in rural households (24.1 per 1,000 live births) compared to urban households (18.1 per 1,000 live births) (Fig 5).

**NMR variations across diverse education groups, 0–5 years preceding the 2016 and 2022 NDHSs.** Disparity in NMR between mothers with no education and those with secondary and higher education increased from 2016 to 2022. In 2022, the NMR decreased in mothers with secondary and higher education compared to mothers without education.

Mothers with no education had a higher NMR of 29.3 in 2022 than 25.7 in 2016. The mothers with secondary and higher education had a decrease in NMR from 16.1 per 1,000 live births in 2016 to 11.1 per 1,000 live births in 2022 (Fig 6).

**Trend on maternal service use and NMR between 2016 and 2022**

Fig 7 shows maternal service utilization and NMR trends from 2016 to 2022. The service use indicators covering at least four ANC visits, institutional delivery, and SBA-assisted delivery were increased. Both at least four ANC check-ups and institutional delivery were increased to approximately 80 percent each in 2022 from around 65 percent each in 2016. Similarly, PNC for mothers and newborns for at least one PNC visit within 48 hours of childbirth also increased from 55 percent in 2016–70 percent in 2022. Despite the increase in service utilization indicators, NMR remained stalled over the period (Fig 7).

   **NMR trend by women empowerment between 2016 and 2022.** Fig 8 presents the NMR by women's household decision-making capacity. Women with lower household decision-making capacity had a higher NMR compared to their counterparts in both survey years. Specifically, in 2022, the NMR among women who do not participate in household decisions was 24.5 per 1,000 live births, which is considerably higher than that of women who do participate in household decisions.

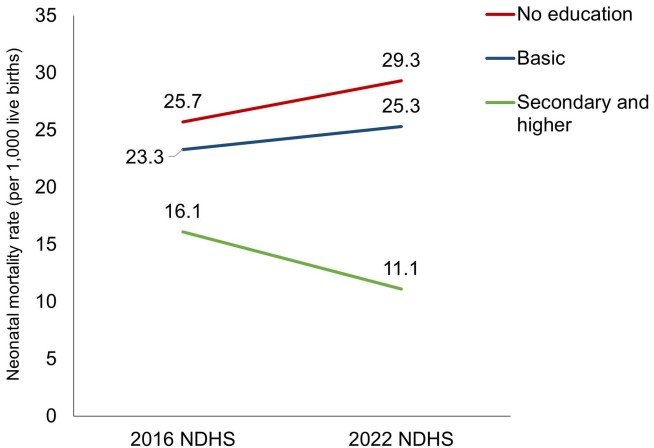

**Fig 6. NMR by education groups, 0–5 years preceding the 2016 and 2022 NDHSs.**

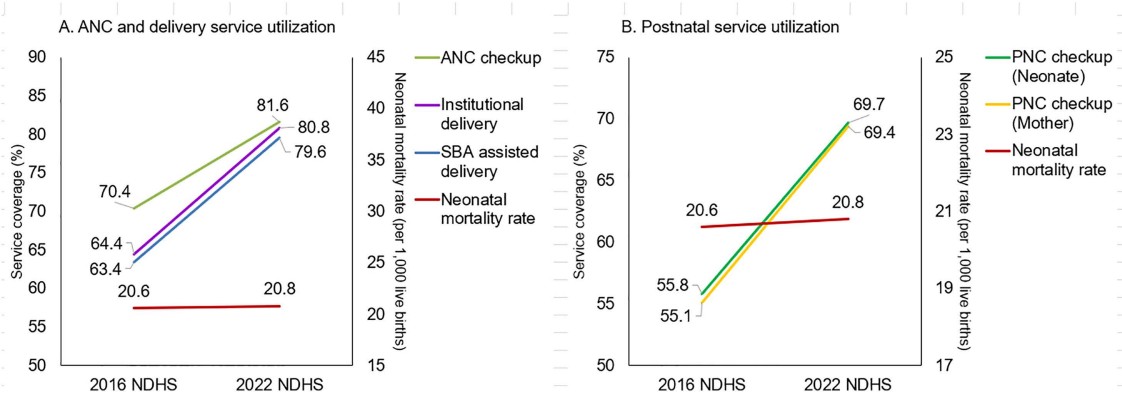

**Fig 7. NMR, 0–5 years preceding the survey and maternal health service utilization, 2016 and 2022 NDHSs.**

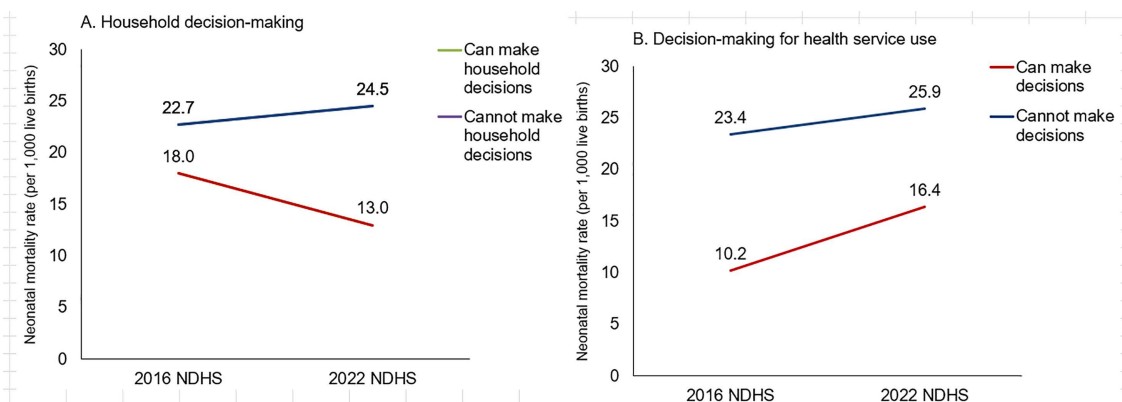

**Fig 8. NMR differentials by women's decision-making capacity for using health services and family planning, 0–5 years preceding the 2016 NDHS and 2022 NDHS.**

The NMR for women who do not make their own decision to use health services in 2022 was higher than those who do: 25.9 percent compared to 16.4 percent, respectively. In 2016, disparity of NMR was more than two-fold higher (23.4 vs. 10.2) among those who cannot make decisions for using health services and family planning (Fig 8).

**NMR differentials across birthweight, 0–5 years preceding the 2016 and 2022 NDHSs.** NMR was higher among newborns with LBWs (<2,500 grams) and whose birthweight was not taken or unknown, compared to normal or large birthweight newborns, for both survey years (Fig 9). In 2022, NMR for higher birthweight newborns were about 7 per 1,000 live births, much lower than newborns with LBWs (24 per 1,000 live births). From 2016 to 2022, NMR for large birthweight newborns declined from 11 per 1,000 live births to 7 per 1,000 live births (Fig 9).

## Determinants of neonatal mortality

Table 2 focuses on variables that exhibit significance at the 0.01 level or better, indicated by two or three asterisks in the second column. These variables effectively differentiate between subpopulations with high and low NMRs. Variables of the eight indicator blocks in the conceptual model (Fig 2) are significantly associated with NMR in the 2022 survey: native language, ethnicity, level of wealth, ecological region, indoor air pollution, maternal education, maternal age, possessing a

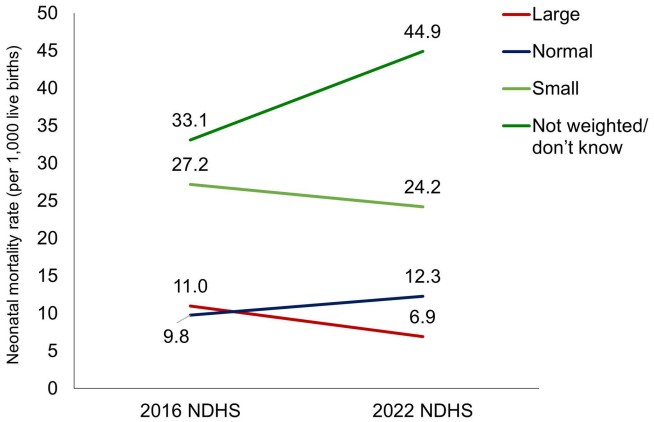

**Fig 9. NMR differentials across birthweight, 0–5 years preceding the 2016 and 2022 NDHSs.**

**Table 2. P-values and pseudo-R² values for covariates' NMR with symbols for the level of significance for the 2022 NDHS.**

| Characteristics | P-value | | Pseudo-R² |
|---|---|---|---|
| **Household characteristics** | | | |
| Mother's language | <0.001 | *** | 0.038 |
| Ethnicity (two categories) | 0.002 | ** | 0.01 |
| Wealth index in terciles | 0.003 | ** | 0.03 |
| Wealth index (poorest and second, middle, fourth and highest) | 0.001 | ** | 0.064 |
| Province | 0.213 | NS | 0.032 |
| Ecological region | 0.034 | * | 0.009 |
| Religion | 0.087 | NS | 0.013 |
| Type of residence | 0.196 | NS | 0.008 |
| Size of household | 0.361 | NS | 0.004 |
| Sex of household head | 0.211 | NS | 0.008 |
| Indoor air pollution | 0.002 | ** | 0.055 |
| Improved water and sanitation | 0.636 | NS | 0.005 |
| **Maternal characteristics** | | | |
| Maternal education | 0.002 | ** | 0.031 |
| Maternal age (five categories) | 0.023 | * | 0.063 |
| Maternal use of tobacco | 0.607 | NS | 0.001 |
| Maternal stature | 0.115 | NS | 0.007 |
| Maternal anaemia | 0.765 | NS | <0.001 |
| Mother ever drinks alcohol | 0.809 | NS | 0.005 |
| **Women's empowerment-related characteristics** | | | |
| Owns mobile phone | 0.153 | NS | 0.004 |
| Possesses a bank account | <0.001 | *** | 0.039 |
| Internet use | 0.001 | *** | 0.016 |
| Empowerment: household decisions | 0.007 | ** | 0.015 |
| Mother's experience of violence | 0.876 | NS | <0.001 |
| Empowerment: health care/family planning decisions | 0.021 | * | 0.014 |
| Newspaper/Magazine | <0.001 | *** | 0.026 |
| Radio/TV | <0.001 | *** | 0.023 |
| **Mother's exposure to health programs** | | | |
| Knows about HMG | 0.972 | NS | <0.001 |
| **Father's characteristics** | | | |
| Father's education | 0.009 | ** | 0.015 |
| Father's occupation (four categories) | 0.015 | * | 0.022 |
| **Birth characteristics** | | | |
| Birthweight taken | <0.001 | *** | 0.078 |
| Sex of child | 0.096 | NS | 0.01 |
| Birthweight | 0.001 | *** | 0.146 |
| Twin birth | 0.535 | NS | <0.001 |
| Perceived birthweight | 0.017 | * | 0.025 |
| **Pregnancy-related characteristics** | | | |
| Birth order | 0.124 | NS | 0.025 |
| Mother's parity | 0.053 | NS | 0.031 |
| Preceding birth interval | 0.001 | *** | 0.054 |
| Wanted last birth | 0.219 | NS | 0.011 |

*(Continued)*

**Table 2.** (Continued)

| Characteristics | P-value | | Pseudo-R² |
|---|---|---|---|
| **Health system-related factors** | | | |
| Time to health facility | 0.205 | NS | 0.009 |
| Birth attendants | 0.010 | ** | 0.033 |
| Place of delivery | 0.052 | NS | 0.052 |
| C-section delivery | 0.106 | NS | 0.008 |
| At least four or more ANC visits | 0.068 | NS | 0.03 |
| Days iron tablets taken | 0.117 | NS | 0.037 |
| Newborn PNC within two days | 0.002 | ** | 0.068 |
| Mother PNC within two days | 0.126 | NS | 0.016 |

Notes: * indicates that the covariate has a significant relationship with the NMR with $p < 0.05$. ** indicates $p < 0.01$ and *** indicates $p < 0.001$. NS indicates not significant. The table presents McKelvey and Zavoina's pseudo-$R^2$.

bank account, using the Internet, empowerment in household decisions, empowerment in health care decisions, exposure to television (TV), exposure to radio, exposure to magazines, husband's education, husband's occupation, birthweight taken, SBAs, and newborn PNC within two days.

However, other covariates expected to influence NMR were not statistically significant: provincial variation, religion, type of residence, size of the household, sex of the household head, improved water and sanitation, maternal use of tobacco, maternal stature, maternal anaemia, mother's experience of violence, having a mobile phone, familiar about mothers' group meeting, time to reach a health facility, twin birth, sex of the child, birth order, and consumption of iron tablet.

**Determinants with high levels of NMR above 25 per 1,000 live births.** Table 3 shows the subpopulations with the highest levels of neonatal mortality in the 2022 survey. Most of them had a high level in the 2016 survey as well. The NMR national average would decline substantially if the rate could be reduced in these subpopulations. A few categories are specific to Nepal, such as the Bhojpuri and Maithili language groups and the Mountain ecological region, but most categories would be found in most countries. These include lower levels of wealth, maternal age, education of the mother or her husband and occupational category. Other categories that stand out are households with indoor air pollution, mothers who do not have a bank account, mothers who have never used the Internet, mothers who do not participate in health care or family planning decisions, and mothers with little exposure to media (newspapers, magazines, radio, or TV).

Several categories in Table 3 are closely linked to potential programs and interventions. High neonatal mortality is observed among children delivered without SBAs, those whose birthweight was not measured or perceived as small, those born within two years of a preceding birth, or those who did not receive PNC within two days (bearing in mind that children who passed away shortly after birth may not have had the opportunity to receive PNC within this timeframe).

**Determinants of low levels (15 per 1,000 live births) of NMR.** Table 4 identifies subpopulations with low neonatal mortality, typically contrasting with the categories of high neonatal mortality in Table 4. These sub populations include the Nepali language group, the Hill ecological region, the highest wealth category, and individuals with secondary or higher education (for both mothers and husbands). Additionally, mothers aged 30–34, those who possess a bank account, have Internet access, are involved in household decisions, read newspapers or magazines, and consume of radio/TV content tend to have lower NMRs. Children also demonstrate higher chances of surviving the neonatal period if they have a normal or large birthweight, had longer preceding birth intervals, were delivered by SBAs, or received PNC care within two days. Caution should be exercised when interpreting low rates, especially below ten, due to wide confidence intervals (see S5 Table).

**Table 3. Categories of the significant (p < 0.05) covariates, in which the NMR is estimated to be 25 or more in the 2016 and 2022 NDHSs.**

| Characteristics | Categories | 2016 NMR | 2022 NMR | Changes | P-value for change | |
|---|---|---|---|---|---|---|
| **Household and sociodemographic characteristics** | | | | | | |
| Respondent's language | Bhojpuri | 25.8 | 43 | 17.1 | <0.001 | *** |
| Respondent's language | Maithili | 23.4 | 27.4 | 4 | <0.001 | *** |
| Wealth index (three categories) | Poorer and poorest | 27.3 | 27.8 | 0.5 | 0.001 | ** |
| Ecological region | Mountain | 36.5 | 26.2 | −10.3 | 0.034 | * |
| Indoor air pollution | Yes | 22.5 | 26.3 | 3.8 | 0.002 | ** |
| **Maternal characteristics** | | | | | | |
| Maternal education | Basic (grades 1–8) | 23.3 | 25.3 | 2 | 0.002 | ** |
| Maternal education | No education | 25.7 | 29.3 | 3.6 | 0.002 | ** |
| Maternal age (five categories) | 15–19 years | 33.2 | 29.3 | −3.9 | 0.023 | * |
| Maternal age (five categories) | 20–24 years | 27.7 | 26.9 | −0.8 | 0.023 | * |
| **Women's empowerment-related characteristics** | | | | | | |
| Possesses a bank account | No | 22.9 | 27.5 | 4.6 | <0.001 | *** |
| Internet use | Never used the Internet | 22.5 | 31.3 | 8.8 | 0.001 | *** |
| Empowerment: health care/family planning decisions | No | 23.4 | 25.9 | 2.5 | 0.021 | * |
| Newspaper/Magazine | Less than once a week | 22.6 | 27.1 | 4.6 | <0.001 | *** |
| Radio/TV | Less than once a week | 22.5 | 26.6 | 4.1 | <0.001 | *** |
| **Husband's characteristics** | | | | | | |
| Husband's education | No education/Do not know | 19.5 | 37.5 | 18 | 0.009 | ** |
| Husband's occupation (four categories) | Manual (skilled/unskilled) | 20.7 | 25.9 | 5.3 | 0.015 | * |
| **Birth-related characteristics** | | | | | | |
| Birthweight taken | Not taken | 33.2 | 44.9 | 11.7 | <0.001 | *** |
| Birthweight | Not weighed or do not know | 33.2 | 44.9 | 11.7 | 0.001 | *** |
| Perceived birthweight | Very small | 35.2 | 37.5 | 2.3 | 0.017 | * |
| Preceding birth interval | ≤2 years | 29.4 | 37.7 | 8.3 | 0.001 | *** |
| Birth attendants | Delivery without SBA | 25.7 | 36.1 | 10.4 | 0.01 | ** |
| Newborn PNC within two days | No PNC | 14.6 | 26.3 | 11.6 | 0.002 | ** |

Notes: * indicates that the covariate has a significant relationship with the NMR with p < 0.05. ** indicates p < 0.01 and *** indicates p < 0.001.

Some variables outlined in the conceptual framework and regarded as potentially important interventions do not emerge as statistically significant. Measures of ANC visits, for instance, do not exhibit statistical significance. It is anticipated that access to and utilization of health services serve as pathways through which household, maternal, and paternal characteristics impact child survival. While the association between ANC care and background variables is explored in Chapter 9 of the final report for both the NDHS 2016 and 2022 surveys [5,6], detailed articulation of these pathways is constrained by low statistical power in this analysis.

## Discussion

This study has shown that since 2000, there has been a modest and uneven reduction of NMR. The 2022 NDHS shows a reduced NMR on the first day of birth when compared to 2016. Equity gaps were seen widened for NMR in 2022 compared to 2016 across household, ethnicity, and wealth index. Over the period of 2016–2021, the paradox of stagnant NMR exists but there has been an increased maternal and neonatal service utilization. Higher neonatal death rate exists in mothers with low or no education and those experiencing early marriage. A positive association of women's

**Table 4. Categories of the significant (p<0.05) covariates, in which the NMR is estimated to be 15 or less in the 2016 and 2022 NDHSs.**

| Characteristics | Categories | 2016 NMR | 2022 NMR | Changes | P-value for change | |
|---|---|---|---|---|---|---|
| **Household and sociodemographic characteristics** | | | | | | |
| Respondent's language | Nepali | 17.4 | 10.3 | −7 | <0.001 | *** |
| Ethnicity (two categories) | Advantaged | 18.3 | 10.7 | −7.6 | 0.002 | ** |
| Wealth index (poorest and second, middle, fourth and highest) | Richer and richest | 16.4 | 8.7 | −7.7 | 0.001 | ** |
| Ecological region | Hill | 16.4 | 13.2 | −3.1 | 0.034 | * |
| Indoor air pollution | No | 17.4 | 11.3 | −6.1 | 0.002 | ** |
| **Maternal characteristics** | | | | | | |
| Maternal education | Secondary and above (≥Grade nine) | 16.1 | 11.2 | −4.9 | 0.002 | ** |
| Maternal age (five categories) | 30–34 years | 15 | 6.8 | −8.2 | 0.023 | * |
| **Women empowerment-related characteristics** | | | | | | |
| Possesses a bank account | Yes | 17.2 | 9.6 | −7.6 | <0.001 | *** |
| Internet use | Used at some time | 13.7 | 14.6 | 0.9 | 0.001 | *** |
| Empowerment: household decisions | Yes, can make decisions | 18 | 13 | −5 | 0.007 | ** |
| Newspaper/Magazine | At least once a week | 19.4 | 11.9 | −7.6 | <0.001 | *** |
| Radio/TV | At least once a week | 19.5 | 12.2 | −7.3 | <0.001 | *** |
| **Husband's characteristics** | | | | | | |
| Husband's education | Secondary and above (≥Grade nine) | 18.2 | 14.1 | −4.1 | 0.009 | ** |
| Husband's occupation (four categories) | Agriculture | 30.6 | 12.5 | −18.1 | 0.015 | * |
| Birthweight taken | Yes, taken | 12.5 | 12.3 | −0.3 | <0.001 | *** |
| **Birth-related characteristics** | | | | | | |
| Birthweight | Large (≥3,500 g) | 11 | 6.9 | −4.1 | 0.001 | *** |
| Birthweight | Normal (2,500–3,500 g) | 9.8 | 12.4 | 2.6 | 0.001 | *** |
| Perceived birthweight | Larger than average | 32.2 | 11.2 | −21 | 0.017 | * |
| Perceived birthweight | Smaller than average | 34.9 | 12.5 | −22.4 | 0.017 | * |
| Preceding birth interval | >Two years | 14.6 | 11.5 | −3.2 | 0.001 | *** |
| Birth attendants | Delivery with SBA | 16.9 | 14.3 | −2.5 | 0.01 | ** |
| Newborn PNC within two days | Yes | 7 | 7.7 | 0.7 | 0.002 | ** |

Notes: * indicates that the covariate has a significant relationship with the NMR with p<0.05. ** indicates p<0.01 and *** indicates p<0.001.

decision-making with lower NMR exists. A positive association of SBA-assisted delivery and PNC newborn with lower NMR exists. The birth interval is lower than two years ago and LBW is associated with higher NMR.

## Modest and uneven reduction of NMR since 2000

The trend analysis shows that the reduction of Nepal's NMR was modest but uneven since approximately 1996, with stagnant rates between two inter-survey periods, 2006–2011 and 2016–2022. Nepal substantially reduced NMR from the inter-survey period, 2011–2016. This impressive reduction might have been due to the expansion of maternal and neonatal services, including basic and emergency obstetric neonatal care services; the Aama program, which provisioned free delivery services and incentives for mothers completing four ANC visits; capacity strengthening of health workers through the community based IMNCI program; and expansion of newborn care services. Reasons behind stagnant NMR since 2016 may be due to the impact of federalism since 2017, when the unitary government was transformed into three tiers of

government, i.e., federal, provincial and local levels and changes happened in the health care delivery system [24]. The transition of federalism required a rigorous staff adjustment process, devolution of the responsibility for basic health care services toward local governments, and changes in the district health system to a municipal health system. Basic health services are under the local government's jurisdiction in Nepal. During the transition, the health system suffered from inadequate staff and, hence, the capacity of the municipalities to provide technical support to the health facilities [25,26].

### Reduced NMR on the first day of birth in 2022 NDHS compared to 2016

The NMR on the first day of birth was lower in the five years preceding the 2022 NDHS survey than that of the 2016 survey. Newborn deaths in the health facility might have been reduced as the country implemented the continuum of care guideline prioritizing postnatal check-ups with a provision to keep new mothers and babies at the health facility at least 24 hours after birth [27]. Another reason could be increased institutional delivery-assisted by SBA, as shown by Nepal service coverage data [28]. In addition, service availability and readiness for delivery and newborn care in 2021 have slightly improved since 2015, as shown by the analysis of health facility surveys in Nepal that may have contributed to reducing neonatal deaths on the first day of birth [29].

### Widened NMR equity gaps across background variables of mothers in 2022

The findings show that the NMR equity gap is evident across household and sociodemographic characteristics. The NMR gap between the poorest and richest households widened in 2022 compared with 2016. With the increased trends of NMR among the poorest households, it would be difficult to achieve the target of NENAP to reduce NMR to 12 per 1,000 live-births by 2030 in the poorest wealth quintile. With the NMR levels of 2016, an equity analysis has projected that the poorest households may only achieve SDG target to reduce NMR to 12 per 1,000 live births by 2067 instead of 2030 [30]. The NMR has increased significantly in mothers whose native languages are Maithili or Bhojpuri, primarily spoken in Madhesh, poorest households, and households with indoor air pollution. The health facility survey analysis for NHFS 2021 has also shown that Madhesh Province has comparatively poor service availability and readiness for delivery and newborn care.

While comparing the NMR across seven provinces, Gandaki Province has already achieved an NMR lower than the SDG target 3.2 (12 per 1,000 live births), whereas Karnali and Madhesh provinces had an NMR of 27 per 1,000 live births for the period of ten years preceding the 2022 survey. The NMR in the Mountain region was significantly reduced from 2016 to 2022, although the rate was higher than the national level. Surprisingly, changes in NMR according to the type of residence were unexpected, although the difference is not statistically significant; the rural NMR was slightly reduced, and urban NMR was slightly increased between the 2016 and 2022 surveys. On the contrary, Nepali native speakers, those from advantaged ethnicity/caste, richest households, and the hilly region had NMRs less than 13 in the 2022 survey, which is close to the SDG target 3.2.

The gap on NMR widened between mothers with no education and those with secondary and higher levels of education in 2022 compared to 2016. Mothers who experienced multiple forms of disadvantages covering lack of education, from disadvantaged ethnicity, and of lower wealth status were less likely to achieve MNH continuum [31]. Mothers with a basic level of education (1–8 grades) and those with no education are associated with significantly higher mortality in the 2022 survey. The results follow similar findings from the previous analyses of 2006, 2011, and 2016 NDHSs, which identified that mothers with no education had higher neonatal mortality [19,32].

### Paradox of stagnant NMR and increased maternal health service utilization in recent surveys

Mothers who received at least four ANC visits also increased institutional delivery by above 80 percent, and SBA-assisted delivery. However, the PNC coverage for mothers and newborns, according to the protocol, was low [28]. Simultaneously, measures of ANC with four or more visits did not appear to be statistically significant with NMR. This may be due to a lack of completion of continuum of care covering institutional delivery and PNC among the mothers who utilized ANC services.

A lack of positive input and process level factors for ANC, such as lack of infrastructure, the provision of commodities and supplies, and health workforce, may have compromised the quality of ANC [33]. A Nepal-based cross-sectional study has shown that the continuum of care completion rate was only around 41 percent [34]. In addition, socioeconomically disadvantaged population have relatively lower utilization of ANC and PNC service in distant part of the country [35].

### Higher neonatal deaths in mothers with low or no education, and those experiencing early marriage

Maternal and paternal characteristics, such as mothers and husband's education, were important determinants for higher neonatal mortality in 2022. A lack of maternal education and young maternal ages of 15–19 and 20–23 years were correlated with higher neonatal mortality. Women with no education or whose husband has no formal education, or whose occupation is manual/skilled labor, also experienced significantly higher neonatal mortality. Adolescent pregnancy may be attributed to higher neonatal mortality, as a hospital-based study has shown that such pregnancy had higher odds of preterm birth and malformation [36].

### Positive association of women's decision-making and use of media with lower NMR

This study also revealed that women's empowerment was significantly associated with lower NMR. Mothers who possessed a bank account and used the Internet had significantly lower NMR in the 2022 survey. Using media and technology and possessing a bank account influenced the use of skill delivery in Nepal [37]. Nevertheless, a lack of household decision-making and decisions regarding health services exhibited a significantly higher NMR. A meta-analysis conducted in low- and middle-income countries shows that a low level of woman's empowerment was associated with higher neonatal mortality [38]. Maternal exposure to the media to obtain health-related messages, such as reading the newspaper, listening to the radio, and watching TV, was also associated with significantly lower NMR. However, familiarity with the Health Mother's Group (HeMG) meeting and attending HeMG did not show significant changes in the NMR over six years from the 2016 survey. The decision to participate in HeMG might be one of the challenges for women when the decision comes from the their husband alone or her family members. [39] In this analysis, an NMR level was not significantly associated with the violence experienced by women. This could be due to the smaller subsample of women selected to participate in the domestic violence questionnaires and the tendency of women to underreport the instances of violence in face-to-face interviews [40].

### Positive association of SBA-assisted delivery and PNC newborn with lower NMR

While examining the determinants, this analysis found that quality of MNH care provided by SBA resulted in significantly lower NMR. Neonatal mortality was significantly higher in babies who were delivered without SBAs or who did not receive PNC for newborns within two days. Mothers may have missed the opportunities for safe delivery and essential newborn care as they lacked service from a SBA [37]. Although statistically insignificant, more newborn deaths occurred at home delivery more often compared to delivery in government or private health facilities. Babies who were delivered at home and lacked SBA-assisted delivery usually did not receive essential newborn care, such as cord care, drying, and wrapping [41].

### Positive association of higher NMR with lower birth interval (less than two years) and LBW baby

Newborns' birth characteristics also showed significant differences in neonatal morality. Newborns with LBW and a small perceived size, and those whose weight were not taken, had higher death rates. The finding of the analysis about LBW is corroborated by the Nepal study, which has shown that premature birth is the major cause of ND [42]. The study team's findings about the higher neonatal mortality among babies born within a two-year birth interval are supported by a meta-analysis, which shows that babies born in short birth intervals were more likely to die in low- and middle-income countries [43]. The study did not show any significant differences in NMR based on the sex of the child.

The study has some limitations. It was acknowledged that a multivariate analysis would be important to test the significance of specific determinants while controlling for the effect of selected background characteristics. However, this was not possible due to neonatal mortality being a relatively rare event in statistical terms. Each survey included only about 100 neonatal deaths in the past five years, which is not enough cases for analysis with multiple independent or control variables. While the study utilized household survey data to examine service use factors, it did not include service facility characteristics, which might have enriched the analysis.

## Conclusions

The trend analysis shows that the NMR are higher in socially and economically disadvantaged groups, which may be a result of poor service utilization. The major health system factors identified were poor service use, such as the use of a SBA at delivery, PNC for newborns, and special care for LBW babies. The equity gap in MNH services needs to be addressed by designing locally led solutions, such as providing MNH messages in local languages, increasing local and female service providers, and targeting the poorest and poorer households, mothers with low education backgrounds, and disadvantaged ethnicity or caste groups. Achieving the SDG target of an NMR of 12 per 1,000 live births will require focused efforts on addressing equity gaps, enhancing maternal education, and improving access to skilled birth attendants at the time of delivery. Further, comprehensive newborn-focused research is needed to identify the underlying causes of neonatal mortality across diverse populations.

## Supporting information

**S1 Table. The numbers of births; the numbers of early neonatal, late neonatal, and neonatal deaths; and the early neonatal, late neonatal, and NMRs per 1,000 births, for all categories of the covariates, for the five years before the 2016 NDHS.**
(DOCX)

**S2 Table. The numbers of births; the numbers of early neonatal, late neonatal, and neonatal deaths; and the early neonatal, late neonatal, and NMRs per 1,000 births, for all categories of the covariates, for the five years before the 2022 NDHS.**
(DOCX)

**S3 Table. The early neonatal rates in the 2016 and 2022 NDHSs, and the difference between them with confidence intervals for the rates and the difference and P-values for the difference.**
(DOCX)

**S4 Table. The late neonatal rates in the 2016 and 2022 NDHSs and the difference between them, with confidence intervals for the rates and the difference and P-values for the difference.**
(DOCX)

**S5 Table. The neonatal rates in the 2016 and 2022 NDHSs and the difference between them, with confidence intervals for the rates and the difference and P-values for the difference.**
(DOCX)

**S6 Table. The p-values for the differences between the 2016 and 2022 NDHSs in the early neonatal, late neonatal, and NMRs for all categories of the covariates, with symbols for the level of significance.**
(DOCX)

**S7 Table. Study variables included in the analysis for the 2022 NDHS.**
(DOCX)

## Acknowledgments

The authors gratefully acknowledge the inputs and guidance for this entire work from USAID Nepal, Ministry of Health and Population Nepal, and ICF, USA.

## Author contributions

**Conceptualization:** Khem Narayan Pokhrel, Resham Khatri, Gauri Pradhan, Tulsi Ram Thapa, Thomas Pullum, Fern Greenwell.

**Data curation:** Khem Narayan Pokhrel, Resham Khatri, Suman Sapkota, Kalpana Gaulee Pokhrel, Tulsi Ram Thapa, Fern Greenwell.

**Formal analysis:** Khem Narayan Pokhrel, Suman Sapkota, Kalpana Gaulee Pokhrel, Tulsi Ram Thapa, Thomas Pullum, Fern Greenwell.

**Methodology:** Resham Khatri, Suman Sapkota, Kalpana Gaulee Pokhrel, Gauri Pradhan, Tulsi Ram Thapa, Thomas Pullum, Fern Greenwell.

**Resources:** Kalpana Gaulee Pokhrel, Gauri Pradhan.

**Supervision:** Resham Khatri, Gauri Pradhan, Tulsi Ram Thapa, Thomas Pullum, Fern Greenwell.

**Validation:** Resham Khatri, Kalpana Gaulee Pokhrel, Gauri Pradhan, Kamal Prasad Chapain, Thomas Pullum, Fern Greenwell.

**Visualization:** Khem Narayan Pokhrel, Suman Sapkota, Kamal Prasad Chapain.

**Writing – original draft:** Khem Narayan Pokhrel.

**Writing – review & editing:** Resham Khatri, Suman Sapkota, Kalpana Gaulee Pokhrel, Gauri Pradhan, Tulsi Ram Thapa, Kamal Prasad Chapain, Thomas Pullum, Fern Greenwell.

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
