## [Decision Letter · Decision Letter 0]

26 Jun 2025

PONE-D-25-19626Elucidating Trends and Underlying Drivers of Neonatal Mortality Stagnation in Nepal: An Analytical Perspective on the 2016 and 2022 Demographic and Health SurveysPLOS ONE

Dear Dr. Pokhrel,

Thank you for submitting your manuscript to PLOS ONE. After careful consideration, we feel that it has merit but we request minor revisions. Therefore, we invite you to submit a revised version of the manuscript that addresses the points raised during the review process.

Reviewer 1:

This work studied trends and the key determinants of the NMR (Neonatal Mortality Rate) in Nepal, using NDHS 2016 and NDHS 2022 dat. data (Nepal Demographic and Health Surveys). It has great importance for the medical and public health fields and can contribute to medical and governmental decision-making.About 43 independent variables were analysed and organized in 8 groups.

The authors collected and analysed all data using apropriated methology. They also applied appropriate methodology for analysis and discussion and reflection on the data. They also addressed that the data show the need for adaptation of medical and hospital practices, as well as public health.

Reviewer 2:

This is a very important paper highlighting the trends, status, and determinants of neonatal deaths in Nepal.

Detailed Comments:

1. Abbreviations and Punctuation:

o Revisit the use of abbreviations throughout the document to ensure consistency and clarity.

o Review punctuation for correctness and coherence across the text.

2. Abstract and Conclusion:

o In the last line, suggest adding that in addition to exploring determinants, rigorous studies on the causes of neonatal deaths are necessary.

3. Page 1:

o In the relevant paragraph, suggest including “2030” to indicate the Sustainable Development Goal (SDG) target year.

4. Page 2:

o Expand MNH (Maternal and Newborn Health) at its first use.

5. Page 3:

o Expand the following abbreviations at first use:

SDIP

CB-NCP

IMNCI

6. Page 4 – Methods Section:

o After the sentence: “The NDHSs is a nationally representative household.”, suggest adding a reference:

“Detailed information for the household and health facility survey sample designs, sample selection, and sample weighting are available in the final reports.”

7. Page 5 – Study Variables:

o The sentence “The NMR is an indicator published in the NDHS final reports and on STATcompiler. [22]” is unclear. Suggest rephrasing for clarity.

o Clarify whether neonatal mortality refers to deaths within 30 days or 28 days. The sentence suggests both:

“Both of these sources use the standard DHS calculation for NMR based on neonatal deaths in the first month of life, or 30 days.” — verify accuracy.

8. Page 7 – Ethical Considerations:

o Clarify whether a separate ethical approval was obtained. If not, suggest writing that the NDHSs obtained ethical approvals from the NHR and a separate approval was not sought for this study based on secondary analysis of the NDHSs.

9. Page 8:

o Expand ND (Neonatal Deaths) at first use.

10. Page 14:

• The sentence is unclear:

“While the association between ANC care and background variables is explored in Chapter 9 of the final report for both the 2016 and 2022 surveys, detailed articulation of these pathways is constrained by low statistical power in this analysis.”

o Which report and which chapter are being referenced? Please clarify.

11. Page 16:

• Consider simplifying the description of federalism. For example, clarify where municipalities belong within the three tiers (federal, provincial, local).

12. Page 17:

• Recommend comparing NMR with the SDG 2030 target, rather than the NeNAP target, for consistency and relevance to international readers.

13. Page 19:

• The phrase “maternal and husband” should likely be revised to “maternal and paternal.”

14. Page 20:

• Expand HeMG (Health Mothers’ Group) at first use.

15. Conclusions and Key Recommendations:

• This section reads like Conclusions only. Recommendations can go into the Discussion section. Suggest strengthening this section by linking conclusions more explicitly to SDG targets.

16. Study Limitations:

• Recommend including a brief discussion of the limitations of the study.

17. Results and Discussion:

• Consider shortening these sections to maintain the reader’s engagement.

18. Table 2:

• Suggest replacing the term “respondent” with more appropriate terminology.

• Clarify if “husband” refers to “father” in the context of the study.

19. Figures:

• Figures 1 and 2 are blurred and difficult to read — suggest improving image resolution.

• Figure 3: Confirm if the “Late Neonatal Period” is 0–28 days or 0–27 days. Please verify.

• Ensure consistent font size and type across all figures and graphs.

We look forward to receiving your revised manuscript.

Kind regards,

Sabita Tuladhar

Academic Editor

PLOS ONE

Journal Requirements:

**Additional Editor Comments:**

Reviewer 1:

This work studied trends and the key determinants of the NMR (Neonatal Mortality Rate) in Nepal, using NDHS 2016 and NDHS 2022 dat. data (Nepal Demographic and Health Surveys). It has great importance for the medical and public health fields and can contribute to medical and governmental decision-making.About 43 independent variables were analysed and organized in 8 groups.

The authors collected and analysed all data using apropriated methology. They also applied appropriate methodology for analysis and discussion and reflection on the data. They also addressed that the data show the need for adaptation of medical and hospital practices, as well as public health.

Reviewer 2:

This is a very important paper highlighting the trends, status, and determinants of neonatal deaths in Nepal.

Detailed Comments:

1. Abbreviations and Punctuation:

o Revisit the use of abbreviations throughout the document to ensure consistency and clarity.

o Review punctuation for correctness and coherence across the text.

2. Abstract and Conclusion:

o In the last line, suggest adding that in addition to exploring determinants, rigorous studies on the causes of neonatal deaths are necessary.

3. Page 1:

o In the relevant paragraph, suggest including “2030” to indicate the Sustainable Development Goal (SDG) target year.

4. Page 2:

o Expand MNH (Maternal and Newborn Health) at its first use.

5. Page 3:

o Expand the following abbreviations at first use:

SDIP

CB-NCP

IMNCI

6. Page 4 – Methods Section:

o After the sentence: “The NDHSs is a nationally representative household.”, suggest adding a reference:

“Detailed information for the household and health facility survey sample designs, sample selection, and sample weighting are available in the final reports.”

7. Page 5 – Study Variables:

o The sentence “The NMR is an indicator published in the NDHS final reports and on STATcompiler. [22]” is unclear. Suggest rephrasing for clarity.

o Clarify whether neonatal mortality refers to deaths within 30 days or 28 days. The sentence suggests both:

“Both of these sources use the standard DHS calculation for NMR based on neonatal deaths in the first month of life, or 30 days.” — verify accuracy.

8. Page 7 – Ethical Considerations:

o Clarify whether a separate ethical approval was obtained. If not, suggest writing that the NDHSs obtained ethical approvals from the NHR and a separate approval was not sought for this study based on secondary analysis of the NDHSs.

9. Page 8:

o Expand ND (Neonatal Deaths) at first use.

10. Page 14:

• The sentence is unclear:

“While the association between ANC care and background variables is explored in Chapter 9 of the final report for both the 2016 and 2022 surveys, detailed articulation of these pathways is constrained by low statistical power in this analysis.”

o Which report and which chapter are being referenced? Please clarify.

11. Page 16:

• Consider simplifying the description of federalism. For example, clarify where municipalities belong within the three tiers (federal, provincial, local).

12. Page 17:

• Recommend comparing NMR with the SDG 2030 target, rather than the NeNAP target, for consistency and relevance to international readers.

13. Page 19:

• The phrase “maternal and husband” should likely be revised to “maternal and paternal.”

14. Page 20:

• Expand HeMG (Health Mothers’ Group) at first use.

15. Conclusions and Key Recommendations:

• This section reads like Conclusions only. Recommendations can go into the Discussion section. Suggest strengthening this section by linking conclusions more explicitly to SDG targets.

16. Study Limitations:

• Recommend including a brief discussion of the limitations of the study.

17. Results and Discussion:

• Consider shortening these sections to maintain the reader’s engagement.

18. Table 2:

• Suggest replacing the term “respondent” with more appropriate terminology.

• Clarify if “husband” refers to “father” in the context of the study.

19. Figures:

• Figures 1 and 2 are blurred and difficult to read — suggest improving image resolution.

• Figure 3: Confirm if the “Late Neonatal Period” is 0–28 days or 0–27 days. Please verify.

• Ensure consistent font size and type across all figures and graphs.

Reviewers' comments:

Reviewer's Responses to Questions

**Comments to the Author**

1. Is the manuscript technically sound, and do the data support the conclusions?

Reviewer #1: Yes

2. Has the statistical analysis been performed appropriately and rigorously? 

Reviewer #1: Yes

3. Have the authors made all data underlying the findings in their manuscript fully available?

Reviewer #1: Yes

4. Is the manuscript presented in an intelligible fashion and written in standard English?

Reviewer #1: Yes

5. Review Comments to the Author

Reviewer #1: This work studied trends and the key determinants of the NMR (Neonatal Mortality Rate) in Nepal, using NDHS 2016 and NDHS 2022 dat. data (Nepal Demographic and Health Surveys). It has great importance for the medical and public health fields and can contribute to medical and governmental decision-making.About 43 independent variables were analysed and organized in 8 groups.

The authors collected and analysed all data using apropriated methology. They also applied appropriate methodology for analysis and discussion and reflection on the data. They also addressed that the data show the need for adaptation of medical and hospital practices, as well as public health.

6. PLOS authors have the option to publish the peer review history of their article (what does this mean? ). If published, this will include your full peer review and any attached files.

**Do you want your identity to be public for this peer review?** For information about this choice, including consent withdrawal, please see our Privacy Policy .

Reviewer #1: No

---

## [Author Response · Author response to Decision Letter 1]

18 Jul 2025

Dear editor and reviewers, kindly find the responses in the separate attached file. We have revised the manuscript addressing your suggestions and feedback. Both track-changed and clean version of the manuscript is attached.

---

## [Editor Report · Decision Letter 1]

6 Aug 2025

Elucidating Trends and Underlying Drivers of Neonatal Mortality Stagnation in Nepal: An Analytical Perspective on the 2016 and 2022 Demographic and Health Surveys

PONE-D-25-19626R1

Dear Dr. Pokhrel,

We’re pleased to inform you that your manuscript has been judged scientifically suitable for publication and will be formally accepted for publication once it meets all outstanding technical requirements.

Kind regards,

Sabita Tuladhar

Academic Editor

PLOS ONE

---

## [Editor Report · Acceptance letter]

PONE-D-25-19626R1

PLOS ONE

Dear Dr. Pokhrel,

I'm pleased to inform you that your manuscript has been deemed suitable for publication in PLOS ONE. Congratulations! Your manuscript is now being handed over to our production team.

Kind regards,

on behalf of

Dr. Sabita Tuladhar

Academic Editor

PLOS ONE